# Depression Detection Based on Hybrid Deep Learning SSCL Framework Using Self-Attention Mechanism: An Application to Social Networking Data

**DOI:** 10.3390/s22249775

**Published:** 2022-12-13

**Authors:** Aleena Nadeem, Muhammad Naveed, Muhammad Islam Satti, Hammad Afzal, Tanveer Ahmad, Ki-Il Kim

**Affiliations:** 1Department of Computer Software Engineering, National University of Sciences and Technology, Islamabad 44000, Pakistan; 2Department of Computer Science, Shaheed Zulfikar Ali Bhutto Institute of Science and Technology, Islamabad 44000, Pakistan; 3Department of Computer Science, Faculty of Computing, Riphah International University, Islamabad 46000, Pakistan; 4Innovation Education and Research Center for On-Device AI Software (Bk21), Department of Computer Science and Engineering, Chungnam National University, Daejeon 34134, Republic of Korea; 5Department of Computer Science and Engineering, Chungnam National University, Daejeon 34134, Republic of Korea

**Keywords:** depression detection, implicit depressive tweets, ternary classification, self-attention mechanism, deep learning

## Abstract

In today’s world, mental health diseases have become highly prevalent, and depression is one of the mental health problems that has become widespread. According to WHO reports, depression is the second-leading cause of the global burden of diseases. In the proliferation of such issues, social media has proven to be a great platform for people to express themselves. Thus, a user’s social media can speak a great deal about his/her emotional state and mental health. Considering the high pervasiveness of the disease, this paper presents a novel framework for depression detection from textual data, employing Natural Language Processing and deep learning techniques. For this purpose, a dataset consisting of tweets was created, which were then manually annotated by the domain experts to capture the implicit and explicit depression context. Two variations of the dataset were created, on having binary and one ternary labels, respectively. Ultimately, a deep-learning-based hybrid Sequence, Semantic, Context Learning (SSCL) classification framework with a self-attention mechanism is proposed that utilizes GloVe (pre-trained word embeddings) for feature extraction; LSTM and CNN were used to capture the sequence and semantics of tweets; finally, the GRUs and self-attention mechanism were used, which focus on contextual and implicit information in the tweets. The framework outperformed the existing techniques in detecting the explicit and implicit context, with an accuracy of 97.4 for binary labeled data and 82.9 for ternary labeled data. We further tested our proposed SSCL framework on unseen data (random tweets), for which an F1-score of 94.4 was achieved. Furthermore, in order to showcase the strengths of the proposed framework, we validated it on the “News Headline Data set” for sarcasm detection, considering a dataset from a different domain. It also outmatched the performance of existing techniques in cross-domain validation.

## 1. Introduction

A study conducted by the WHO in the year 2021 states that depression is the second-most common cause of the global burden of diseases. In today’s world, depression is one of the leading causes of disability across the globe, as it is the most-common and detrimental mental disorder affecting the population all over the world [1]. Mental illness in the form of depression is so widespread that its global prevalence is estimated to be between 6 and 21 percent [2]. The situation of mental illnesses, particularly in developing and underdeveloped countries, where healthcare resources are scarce, is even worse. The presence of depression in the young population is estimated to be 5 percent across the world and 20 percent in its milder forms (i.e., mild depression, partial symptoms, and probable depression) [3]. The provision of mental health care is infrequent; thus, often depression and other mental disorders remain untreated. Sometimes, the disease is not even diagnosed or identified.

The most-important part of the treatment of all diseases, particularly mental disorders including depression, is the identification of symptoms of depression among patients. The detection of depressive symptoms well before its assessment and treatment can significantly improve the likelihood of the reduction of depressive indications and underlying diseases. Moreover, it can alleviate negative implications for health and well-being, as well as social, economic, and social life [4]. The potential diagnostic techniques are often expensive for the poor populations of underdeveloped countries and involve the patient reporting to a mental health care practitioner, and patients or caregivers are required to explain the symptoms. This aspect of diagnostic techniques often leads to non-identification of depression in patients and results in untreated individuals. There is a need to develop a diagnostic method without the physical involvement of the patient at a reduced or no financial cost.

Over the last decade, social media has emerged to be an extensively used platform for the exchange of ideas and information. A small chunk of text can echo the mental state of a person. Hence, practitioners can gain a good amount of intuition about the mental well-being and health of an individual from a Facebook post, a tweet, or an Instagram post [5,6]. There is an explosive amount of literature that caters to the part of social media on the anatomy of social dealings such the break-up of relationships, smoking and drinking reversion, suicide celebration, sexual harassment, and mental illnesses [6,7]. The ever-growing increase in the use of social media across the globe provides the opportunity to use social media as a tool for depression detection [8].

Referring to the diagnostic techniques required for depression detection, many studies have experimented with various techniques and suggested numerous frameworks. The techniques make use of different forms of data based on social-media-based depression datasets, multi-modal datasets (audio and video interviews), survey-based datasets, and EEG-signal-based depression datasets. Studies that have used social-media-based depression datasets have mostly resorted to Facebook, Twitter, Reddit, Reachout.com, Instagram data, and some other social channels. Most of the studies focus on explicit depression detection in which there is a clear statement regarding the disease [7,8,9,10]. Moreover, with explicit detection, the time factor of social media usage has also been considered. Depressed patients usually show the tendency of insomnia, so they stay awake at night and use social media more frequently [8,11,12,13,14]. In this task, SVM [7,8,15] has shown great success with the highest accuracy of 85 percent and Convolutional Neural Networks (CNNs) [12,14] have shown good results with the highest accuracy of 88 percent. However, an explicit statement of depression and the time factor are not the ultimate measures to rely upon, and our goal is to detect a sarcastic tone or implicit statement about the disease. A person may not be stating it clearly, but can still be even suicidal.

Contrary to social media data, where a person is openly expressing thoughts, multi-modal depression datasets, which contain audio and video interviews of patients, have also been used in various studies along with machine learning and deep learning models [16,17,18,19,20,21,22,23,24]. The shortcoming of these methods of detecting through multi-modal datasets is that they involve patients directly. In developing countries, where resources are scarce as ratio of the population [4,8], patients cannot indulge in such long interactions with doctors. Moreover, it becomes a difficult and expensive method to analyze audio and video. In such methods, CNNs again have proven to be highly efficient because of being proficient in image/video processing [7,15,20,22]. Furthermore, the survey-based depression datasets have the same issue of involving the patient and obtaining biased answers [19,20,21,22]. In addition to that, EEG-signal-based datasets have also been employed in studies and involved image processing [25,26,27].

For the stated task, we resorted to social media data and focused on the problem of the implicit depression detection task. In the studies that were surveyed, there is no facilitation of implicit depression categorization. The datasets are even annotated on an “explicit” basis, where a patient states that he/she is depressed. Moreover, there is no study on the detection of first-person (I, we), second-person (you, they), and third-person (he, she, my mother/father, etc.) depression. This paper proposes the exploration of publicly available Twitter depression datasets through Natural Language Processing and involves the manual annotation of data into binary and ternary labels, where binary labels are for first-person depression and ternary labels are for second- and third-person depression, Additionally, machine learning and deep learning algorithms were applied for the classification task, along with various feature extraction techniques and pretrained word embeddings.

Ultimately, a novel hybrid deep learning SSCL framework with an attention mechanism is presented. The SSCL model consists of LSTM and CNN layers in order to boost the depression detection classification task along with a GRU and self-attention mechanism, to capture the contextual information. Overall, models employing the GRU have proven to show better efficiency and effectiveness as compared to other Recurrent Neural Networks (RNNs) such as the Long Short-Term Memory (LSTM) model [9] alone. Moreover, to boost the efficiency of the model, an attention layer was added to enhance capturing of contextual/implicit information in a tweet and performed the task of depression detection with an improved accuracy and F1-score as compared to other novel models. In addition to that, the experimental results attained on a real-world/unseen tweet dataset showed the dominance of the proposed SSCL method as compared to the baseline methods. The pivotal contributions of this study are stated as follows:A hybrid deep learning model is proposed, which uses a self-attention mechanism focusing on the sequential, long-distance relationship of words, as well as their contextual information, by considering the weights of the words.An existing benchmark dataset was annotated with the help of domain experts, to discriminate between the implicit and explicit nature of expression in tweets. This improved the accuracy by reducing the false positive and false negative rates.The dataset was further annotated to formulate the problem of depression detection into binary and ternary classification tasks, catering to first-, second-, as well as third-person depression mentioned in tweets.The proposed framework was evaluated and shown to work well on unseen data (random tweets, other than in the training and test dataset splits) and text classification tasks from a different domain (sarcasm detection).

The paper structure is as follows. Section 2 is the related work in which relevant papers are studied. Section 3 explains the materials and methodology for this research. Section 4 presents the experimental results of machine learning and deep learning on raw, as well as modified labeled data. Furthermore, Section 5 gives the results and discussion, elaborating on the analysis of the experimental results. Section 6 concludes the work, and Section 7 discusses the future work possibilities and the directions of the current research.

## 2. Related Work

Depression detection is of great importance as it has increased around us. It is a prevalent mental disorder worldwide that carries huge burdens with it such as suicide, economic burden, societal disturbance, physical damage, and many profound health concerns. Due to the damaging nature of the disease, there has been a good amount of research and literature on it. In this section, we analyze the related work and cover the maximum dimensions of the work and techniques that have been used in the domain of depression detection. The relevant studies that were included use datasets based on social media (Facebook, Twitter, Reddit, and Reachout.com), multi-modal datasets (audio and video), survey-based datasets, and EEG-signal-based depression datasets.

The social-media-based depression datasets have been used in various studies. Studies inculcating datasets from frequently used social media sites, such as Facebook, Twitter, Reddit, and Reachout.com, have been explored. Uddin et al. used texts depicting depression symptoms. They used a Long Short-Term Memory (LSTM) Recurrent Neural Network (RNN) to classify texts of self-perceived symptoms of depression. The symptoms were predefined by medical professionals. The model was used to discriminate texts depicting depression symptoms from posts with no such descriptions. Finally, the trained RNN was used to automatically predict depression posts. This RNN along with LSTM gave a mean accuracy of 98 percent, but did not cater to explicit depression detection [9].

The study [10] used the textual content of the tweets of users and analyzed the semantic context in texts through deep learning models. The proposed model was a hybrid of two deep learning algorithms, a Convolutional Neural Network (CNN) and Bi-directional Long Short-Term Memory (BiLSTM). After optimization, it gave an accuracy of 94.28 percent. Moreover, in [11], machine-learning- and lexicon- (dictionary and corpus) based approaches were used for text classification in the task of depression detection. The results of balanced and imbalanced data techniques were also explored. Machine learning algorithms were applied, but a hybrid technique provided better results in detecting depression. A hybrid model of LSTM and SVMs was created to detect depression with more accuracy. With data balancing from SMOTE and using LSTM with SVM, the highest accuracy achieved was 83 percent.

The study [12] proposed Multi-aspect Depression Detection with a Hierarchical Attention Network (MDHAN) for detecting depression through the social media posts of users. The task was performed by extracting features from the user behavior and the user’s online timeline (posts). The model consisted of a hybrid setting to tackle user behavior through the MLP and user timeline posts in the HAN to calculate each tweet and word’s importance and capture semantic sequence features from the user timelines (posts). The MDHAN achieved the best performance with 89 percent for the F1-score, to detect depression on Twitter.

Islam et al. used Facebook users’ comments for behavioral examination in [15] and applied basic machine learning algorithms, using primary linguistic features. Moreover, they also considered the timings of Facebook posts and classified the data based on particular words and timings, for which SVM performed the best. In [8], a socially meditated patient portal app was developed for the detection of depression-related markers on Facebook using ML techniques. The observed features were generated from Facebook, and the unobserved features were generated using SMPP. In this study, again, the SVM model resulted in the best average result. Both of the studies used basic ML algorithms, and the classification criteria were explicit rather than implicit.

In another study, the text classification model SS3 for early depression detection was presented in [7], using CLEF 2017—the content posted by users/subjects on Reddit. The model consisted of two phases. In the first phase of classification, a document was split into multiple blocks until the words were reached, and in the second phase, the global value of a word for each category was computed. The technique was novel, but a higher rate of false positives and false negatives could be expected. Another study [13] also employed the same dataset (CLEF-2017). Various feature representations with features such as stylometric and morphology features containing parts-of-speech proportions were used. SVM and random forest were used for the classification of the different features, in which SVM gave the best result. The set of features was limited in this study, and the context could not be captured. Instagram data with images, text, and temporal aspects were researched in [14]. A summed depression score for each post by the static weight and time-adapted weight was calculated. a CNN was used to extract image features, and Word2Vec was used to learn vector representations. The results were based on the score, and again the implicit nature of the content could not be captured.

In [12], a hybrid model was proposed using a Twitter-based dataset. The model used pre-trained word embeddings to encode the words in user posts, and the proposed model attained superior performance by merging a BiGRU with a CNN to detect depression on Twitter. In [18], the Reachout.com (a platform for young people to discuss their everyday issues) dataset was manually labeled based on posts’ urgency (labels: crisis, red, amber, green). Psycho-linguistic features and network features were utilized, in combination, and LSTM was used for anomaly detection. The study categorized depression on the basis of labels and did not include the classification of depressed and non-depressed patients.

Many studies have used multi-modal datasets including different modes of data such as audio, video, and text. DAIC-WOZ, i.e., the “Distress Analysis Interview Corpus”, and AVEC-2019, the “The Audio/Visual Emotion Challenge & Workshop”, were the most commonly used. The study [16] employed speech signal processing with hand-crafted features to apply depression detection techniques using deep learning methods. This paper elaborated on methods for acoustic feature extraction and presented algorithms for classification. A multimodal dataset comprising AVEC2013 and DAIC-WOZ (2014) was used, and the classification task was performed using a CNN and LSTM. Similarly, Reference [17] used audio recordings of the individuals’ voices. The DAIC-WOZ database was used to conduct text analysis on a word level using Natural Language Processing (NLP) techniques and a voice quality analysis model on tense for the breathdimension. The text analysis model showed the best performance with an F1-score equal to 0.8 (0.42) for non-depressed (depressed) individuals, while the voice quality model score was 0.76.

In [28], the model that was presented for the depression detection task consisted of transformers at the top and a 1D-CNN at the bottom. Features from both models were fed to the feedforward network, which classified the depressed and normal people. A graph attention model to handle multi-modal (audio, text, images) data for depression detection was proposed in [29]. A 10-layer temporal CNN was used to obtain the features of multi-modal data and perform classification for the depression detection tasks. In [30], the DAIC-WOZ dataset was used in combination with the AVEC depression dataset. AVEC was used for training purposes, whereas the tuning was performed on DAIC. A novel multi-modal framework with a hybrid deep learning approach combining supervised deep learning, shot learning, and human allied interactions was presented in the study. In [19], again, DAIC-WOZ was used, and a multitask setting was employed to combine regression and classification. The model consisted of a three-layer BGRU model with 128 hidden units and four different text embeddings, namely Word2Vec, FastText, ELMo, and BERT.

Another study [20] explored the AVEC challenge for audio, video, and text descriptors and developed a hybrid classification and estimation framework for depression. A Deep Neural Network (DNN) and a Deep-Convolutional-Neural-Network (DCCN)-based audio and visual multi-modal depression recognition framework was proposed. In particular, teenager psychological stress was studied in [21] by a self-developed mobile app, “Happort”, which collected sleep and exercise data via a wearable wrist sensor. A multimodal interactive fusion method involving text images, sleep, and exercise data was proposed.

Survey-based depression datasets have also been employed by many studies. Survey-based datasets were either collected in the form of self-developed questionnaires, which were filled in by participants, or by using an already available dataset such as *Depression Anxiety & Stress Scale Questionnaire—21 Questions (DASS-21)*. In [24], the data collection was performed through the Amazon Mechanical Turk (MTurk) portal, by using a survey. Afterward, complex and nonlinear relationships between features were developed using machine learning algorithms, namely: KNN, LR, NN, RF, and SVM-Linear. In another study [25], machine learning algorithms were deployed for learning participants who were symptomatic and non-symptomatic for depression, and a differentiation between depression and anxiety was made by sensing a unique pattern of biased responses to emotional inducements, Random forest was used to identify patterns in the behavioral measures. Studies using survey-based datasets require patient involvement, whereas the idea of our research is to detect depression requiring minimum difficulties for the patients. The techniques used in these studies are also basic and have little novelty.

An Electroencephalogram (EEG) is an assessment of the brain by means of electrodes, which are metallic disks attached to the head/scalp. The test senses electrical activity in brain cells and interconnects via electrical impulses. The activity is shown in the form of curvy lines on an EEG recording. The EEG of depressed patients differs from non-depressed patients in terms of brain cell activity. Many studies have used EEG recordings’ datasets to study abnormal activity in depressed patients. In [31,32,33], EEG-based datasets were used. Deep learning models such as CNNs and LSTMs have been employed to perform classification based on EEG signals or images.

For the depression detection task, our approach is different from all the other studies in that we manually annotated the data after a consultation by a group of mental health practitioners to capture the explicit, as well as the implicit context of tweets. Furthermore, after preparing the quality dataset, we used pre-trained Twitter word embeddings to extract the features. Then, various methods of comparative studies were applied, which ave better results on our dataset, and our own hybrid deep learning model performed exceptionally well.

## 3. Materials and Methods

This section includes details about the dataset and its annotation and the methodology that was followed in this research. Our research is divided into two modules, data preparation and data classification. In the first module, we collected the data and manually annotated them as binary and ternary labels and, furthermore, pre-processed them to prepare the data for classification. In the second module, we applied various feature extraction techniques to the prepared data and applied machine learning and deep learning classification algorithms to the binary and ternary data exclusively. Lastly, a hybrid framework is proposed for depression classification. Figure 1 illustrates the modules of our research.

The section is divided into four subsections; Subsection “*Methodological Overview*” sheds light on the generic phases of the research and proposed methodology. Subsection “*Dataset Preparation*” elaborates on the procedure and techniques through which we prepared our dataset. Subsection “*Feature Extraction*” describes the techniques used for feature extraction of the dataset. Lastly, Subsection “*Classification Model Task*” explains the main task of our model.

### 3.1. Methodological Overview

This section elaborates the methodology that was followed in conducting the research. We first collected the depression dataset and observed that the labeling of the dataset was based on the presence of the keyword “depression”. The occurrence of the keyword decided the label to be true for a tweet to be in the depressive category. Hence, the foremost thing we carried out was to manually annotate in order to capture the implicit and explicit nature of depressive and non-depressive tweets. After the data annotation task, Twitter-specific preprocessing was carried out for data cleaning, in which we removed duplicate comments, hyperlinks, hashtags, special characters, punctuation, extra spaces, and other languages. The manual annotation of the data catered to two types of classification tasks, i.e., the binary classification task and the ternary classification task. In binary labeling, we labeled the data such that two categories of labels were created, one for depressive tweets reflecting depression even if it was being implicitly stated and the other for non-depressive tweets. In ternary labeling, three categories were created, one for first-person depression, i.e., the person referring to his/her own depression, one for third-person depression, i.e., the person referring to some other person’s depression, and a third category for non-depressive tweets. Then, we performed the lower-casing of letters and retained the stop words, as they indicate the first-person (I, we, us) depression and third-person depression (he, she, it, them). Moreover, with label correction, we were able to cater to the classification of the implicit and explicit nature of depressive comments. As a part of the preprocessing, we tokenized the data using a spacy tokenizer, to further process the data through feature extraction. Various feature extraction techniques using TF-IDF, N-gram, and pre-trained word embeddings were used. After feature extraction, the classification algorithms were applied on the raw labeled, binary labeled, and ternary labeled data separately. At first, the basic classification algorithms were applied on the raw labeled data. Then, we applied basic ML classifiers on the modified labeling and observed that, on modified labels, they showed better results. Then, we applied various combination and hybrid techniques in order to study the best-fit technique for the problem at hand. The dataset was separated into training and evaluation data in a ratio of 80:20, respectively. We applied supervised machine learning algorithms in combination with TF-IDF and N-grams, to study the outcomes of modified labeled data on the learning process, and we applied the same combinations of machine learning algorithms with TF-IDF and N-grams on the raw labeled data. The modified labels showed better results on real-time tweets, as compared to raw labels. Afterward, with our modified labeled data, deep learning algorithms were explored with the N-gram technique and word embeddings techniques such as FastText, Word2Vec, and GloVe. Ultimately, a novel deep-learning-based hybrid model with an attention mechanism is proposed. In the subsequent subsections, the dataset preparation, Machine Learning (ML) classification algorithms, along with Deep learning algorithms are specified.

All the machine learning and deep learning algorithms were explicitly applied to binary and ternary data, and their results are discussed later in the experimentation section. Figure 2 presents the overall methodological summary of the depression classification task.

### 3.2. Dataset Preparation

The dataset [34] that was employed in this research was acquired from an open-source platform known as Kaggle. The dataset we chose consists of a file containing 31,000 tweets with depressed and non-depressed subjects with binary labels. The problem that was observed in the raw dataset was regarding the labeling of the tweets. Labeling was performed such that the tweets containing keywords or terms related to depression were regarded as depressive and labeled as 1, whereas tweets that did not contain any related keyword or tweets that were off-the-topic were labeled as 0 and regarded as non-depressive tweets. As a result, the depression classification was not accurate due to poor and inaccurate labeling. Furthermore, the dataset also did not handle implicit statements about depression in which a person is not stating clearly the disease, but the tweet or the statement hints at depression. In short, the dataset labeling undertaken was on the basis of keywords related to depression, for instance the tweets containing explicit keywords such as “depressed”, “want to die”, “life is miserable”, etc., were labeled as “1”. The dataset labeling did not cater to implicit comments, but only explicit tweets were observed as depressive tweets.

Consequently, this gave rise to the problem of False Positives (FPs) and False Negatives (FNs) in the dataset. False positives were the tweets that were label “1”’, but actually, they should have been labeled “0”. False negatives were the tweets that were labeled as “0”, but actually should have been labeled “1”. The False Positives (FPs), False Negatives (FNs), True Positives (TPs), and True Negatives (TNs) are elaborated in Table 1. Moreover, Table 2 lists some examples of FPs and FNs. It was necessary to eradicate FPs and FNs because they affect the accuracy negatively, so the labeling was performed such that it minimized the above-mentioned problem to a great extent.

**Dataset hypothesis:** Our hypothesis was to correctly annotate the data so that the model can learn in a way that it can reduce the rate of FP and FN results and increase True Positives (TPs) and True Negatives (TNs), i.e., implicit and explicit result detection. Therefore, a more accurate detection can be performed.

The raw dataset contained tweets labeled as “1” and “0”. The annotation was challenging with respect to the fact that it contained false positive and false negative labels. An experienced mental health practitioner was consulted for the stated purpose, and a criterion was developed for the modification of the annotation. The process was carried out manually to ensure reliability. Table 3 states the existing criteria for data labeling.

Another significant part of the data preprocessing included the cleaning and preparation of the data for further analysis and effective model training. Initially, the raw file contained 31,000 tweets in total, out of which 15,999 were labeled “1” and 15,000 were labeled “0”. The detailed statistics are listed in Table 4. According to the depression detection domain, the prepossessing steps were carried out on data including removing duplicate comments to avoid skewed metrics, and to make the learning process impartial, appending hashtags in the tweet for information extraction, hyperlinks, punctuation, extra spaces, other language, and special characters removal were carried out. Moreover, lower-casing of letters was performed. Stop words were retained as they play a significant role in identifying the first-person (I, me, myself, we, us, our, and ourselves) or third-person depression (she, her, herself, he, him, and himself), so pronouns were retained to make the learning of the model more accurate. Other stop words were removed. Tokenization is an integral part of preprocessing for all ML and DL algorithms. In this research, the “Spacy Tokenizer” was utilized for the segmentation of the text. This is performed on the basis of white spaces between words. Moreover, the tokenizer checks for other symbols such as single inverted commas. For example, “shouldn’t” would be split into “should” and “n’t”. The Spacy tokenizer has been specially built for Natural Language Processing and deep learning, so it was applied to our dataset for tokenization.

Label correction or data annotation was carried out as a part of the preprocessing. Following are the statistics concerning label correction. After the label correction, the number of tweets in every class is stated in Table 4.

### 3.3. Feature Extraction

In this study, the textual features of the tweets were used as the input to the classification models. Feature extraction was performed by using different techniques, mainly the TF-IDF and N-gram techniques. For N-gram, we employed uni-gram, bi-gram, uni–bi-gram, and uni–bi–tri-gram. All of these feature extraction methods are applied with dominant machine learning models.

Term Frequency–Inverse Document Frequency (TF-IDF): This is utilized to quantify a term or a word in the document and computes a weight for every term that specifies the significance of the term in the document and corpus. Each resultant vector consisted of the extracted features of a tweet. N-grams: In this technique, N specifies the number of words in a sequence that are grouped together within a given window and used as a feature. In this study, we used 4 sets of N, i.e., N = 1, 2, 1,2, 1,2,3. Table 5 elaborates the details of the N-gram settings that were as feature extraction methods in this study.

Moreover, for deep learning models, feature extraction has been performed by using pre-trained word embeddings mainly Word2Vec, FastText, and GloVe. These word embeddings have been utilized along with various deep learning models and deep learning ensembles:

*Word2Vec:* Word2Vec is a pre-trained model developed by Google for performing sentiment analysis. This basically merges the effective employment of the skip-gram and continuous bag-of-words architectures for calculating vector representations of words. Google News Vector with a word vector model (3 million 300-dimensional English word vectors) was utilized in this study.

*FastText:* This is another way of learning word representation, developed by Facebook’s AI. FastText is basically a library that is used for the learning of word embeddings and obtaining vector representations of words. In this study, 2 million word vectors trained with subword information with 300 dimensions on Common Crawl-pretrained (600B tokens) FastText English word vectors were used as one of the feature extraction methods.

*GloVe—Global Vector for Word Representation:* Stanford’s GloVe is an unsupervised learning model for finding meaningful vector representations for words. Pretrained word embeddings are already trained on a larger corpus, which can be used to solve the similar task of Natural Language Processing. In this study, GloVe’s pre-trained vector model was used to map the tweets onto the vector space. Twitter’s 2B tweets with 27B tokens and a 1.2 M vocab with 200 dimensional vectors were used in this study for extracting the textual features of tweets.

### 3.4. Classification Model Task

Principally, the task of the classification model is to establish a prediction model for the categorization of depressed and non-depressed tweets. The classification was performed considering the textual features as the input. Our corpus of tweets “C” was divided into a training set and a testing set in the ratio of 80 to 20 percent, respectively.

“Tr” is considered as our Training set and “Te” as our Test set, where “Tr=t1,t2,…,tn”, i.e., consisting of n tweets, such that every tweet is annotated as depressed and non-depressed with a class label.


*The labels are:*

(1)
L=l1|l2orL=l1|l2|l3



All the classifier models were used to perform the two tasks discretely:1.In the binary classification task, the classifier were used to predict the binary classes, i.e., l1 = class label “0” for non-depressed and l2 = class label “1” for depressed tweets.2.In the ternary classification task, the classifier was used to predict the ternary classes, i.e., l1 = class label “0” for non-depressed and l2 = class label “1” for depressed tweets, where a person is referring to his/her own depression. Lastly, l3 = class label “2”, for a person who is referring to somebody else’s depression or providing some information about depression, for example providing the contact for a suicide helpline in a tweet.

The evaluation metrics that were used in this study to gauge the performance of our classifiers were the accuracy and F1-score. Both of the metrics take into account, all four significant dimensions, i.e., True Positives (TPs), True Negatives (TNs), False Positives (FPs), and False Negatives (FNs) (refer to Table 1).

Accuracy: This is the measure of all the cases that have been correctly identified, which means it takes into account all the accurately/correctly classified cases.
(2)$$Accuracy=\frac{TP+TN}{TP+FP+TN+FN}$$

F1-score: The F1-scores takes the precision and recall into account, and it is a result of the harmonic mean of the precision and recall. The F1-score is an improved measure over the accuracy metric because the F1-score measures the cases that have been incorrectly classified.
(3)$$F1-Score=2\times \frac{Recall\times Precision}{Recall+Precision}$$

Precision is the measure of the positive cases that are correctly identified from all the predicted positive cases.
(4)$$Precision=\frac{TP}{TP+FP}$$

Recall is the measure of the positive cases that are correctly identified from all the positive cases existing in the dataset.
(5)$$Recall=\frac{TP}{TP+FN}$$

The accuracy measure is used when all of the classes are likewise significant. Therefore, it can be easily deduced that we use accuracy when true positives and true negatives are equally important and there exists a class balance. However, accuracy is not a durable measure in real life, because class imbalance is common in real-life problems. Therefore, keeping this in consideration, we used the F1-score. The F1-score is basically the harmonic mean of the precision and recall of the model. Thus, the F1-score considers the both false positives and false negatives. Hereafter, it is used when the false negative and false positive case are crucial, and in our case, the wrong detection of depression (FP) and no detection of depression when it exists (FN) are vital information that cannot be lost. Moreover, the F1-score handles class imbalance well, and in our ternary classification problem, the classes are imbalanced. In this study, we present both the accuracy and F1-score as evaluation metrics, but because of the mentioned reasons, the F1-score is an imperative metric as compared to accuracy.

## 4. Experimental Analysis

For the above-mentioned classification tasks, seven prevalent machine learning classifiers were employed to evaluate the performance of each. The ML classification models used were the K-Nearest Neighbors (KNNs), Naïve Bayes (NB), Random Forest (RF), Logistic Regression (LR), Support Vector Machine (SVM), Decision Tree (DT), and Multilayer Perceptron (MLP). Deep learning algorithms were also experimented with, namely: feedforward networks, GRUs, CNN, LSTM, BiLSTM, attention layer mechanism, and ensemble/hybrid models. The subsection *“Machine Learning Classification Model”* presents the experimentation result of raw, binary, and ternary labeled data with the ML algorithms. The subsection *“Deep Learning Classification Model”* presents the experimentation result of binary and ternary labeled data with the DL algorithms.

### 4.1. Experimental Analysis of Machine Learning Classification Models

#### 4.1.1. Binary Classification Task with Machine Learning Classifiers

This section elaborates on the experimental analysis regarding the binary classification task. In this study, all the classifiers were examined for the task of depression detection. This section elaborates on the application of the above-mentioned ML algorithms on the raw labeled data and the problem of over-fitting on raw labels. Afterwards, it discusses the application of all machine learning classifiers on the modified labeled data.

We first tested all the classifiers on the raw labels of the dataset. Through experimental analysis, we observed that the classifiers were overfitting. We used two measures to gauge the performance of the classifiers, i.e., accuracy and F1-score. The highest accuracy observed was 99.9, and the F1-measure was 99.8 for the training and test datasets with raw labels (refer to Table 6). All the classifiers were trained on explicit depression statements, that is where a person has used a particular term, it was labeled as a depressive tweet.

Consequently, there was a problem of overfitting. This occurs when, unfortunately, the algorithm cannot perform precisely and accurately on unseen data. This overthrows the purpose of machine learning algorithms because the generalization of the model to novel and unseen data is the eventual aim. The aim of machine learning algorithms makes us utilize them every day to classify data and make predictions. Because of the noise in the labeling, the model overfit and was unable to perform.

Afterwards, we fed all the classifiers with a small set of unseen or real-time tweets discretely. The classifiers were unable to capture the implicit tweets, which depicted that the person was depressed, for example in the test tweet dataset *“I just want to be rich and not stressed and crying all the time.”* In this tweet, a person is stating indirectly that he/she is depressed, but the classifiers were unable to capture this. Hence, to stop the model from overfitting and capturing the explicit and implicit statements about depression, the labeling was modified.

In this study, seven machine learning classifiers were applied to the dataset after modifying its labels, to prevent overfitting and capturing the explicit and implicit content of the depressive and non-depressive tweets. After the application of the classifiers to the modified labeled dataset, we obtained the training and test accuracy and F1-scores. The models were again evaluated against a small set of real-time tweets/unseen data. The classifiers performed better this time and captured the implicit nature of the tweets also through accurate learning. Our hypothesis was increasing the true positives’ and true negatives’ detection. This was proven by the fact that, after the modification of the labels of the dataset, the training was more accurate. Hence, the model performed well on unseen data in the following terms:1.Decreasing false negatives and false positives;2.Increasing the rate of true positives and true negatives;3.Increasing performance in terms of the accuracy and F1-score on unseen/real-time data.

Table 6 presents the results of all the machine learning classifiers applied on raw labeled binary data, in which implicit tweets were not labeled as depressed, rather being labeled asnon-depressed. The feature extraction techniques used with the classifiers were TF-IDF and N-grams (uni-gram, bi-gram, uni–bi-gram, and uni–bi–tri-gram). The data used for the training and testing of the classifiers were divided on the basis of the 80:20 rule, i.e., 80 percent data for training and 20 percent data for testing.

It is noticeable from Table 6 that SVM and logistic regression performed equally well with all of the feature extraction techniques except bi-grams. With bi-grams, multilayer perceptron gave the best results overall. However, it is evident from the results that the classifiers were overfitting.

In order to evaluate the performances of classifiers on unseen data and establish the fact that the classifiers were overfitting, a set of real-time tweets was tested separately for each model with every feature extraction technique. The real-time data were a small set of tweets that consisted of both implicit and explicit tweets. Table 7 presents the results of the ML classifiers on the real-time tweets dataset. It was observed that the model was unable to capture the implicit statements about depression and was highly overfit. Mostly, the classifiers output the wrong labels for implicit statements for depression, whereas they classified the explicit statements well for the real-time tweets. The classifiers gave diverse results with every feature extraction technique. It was evident that the highest F1-score of SVM was with uni–bi–tri-gram with an F1-score of 76.4.

In order to capture the implicit content and fix the problem of overfitting, the labels were modified and implicit statements about depression were also marked as depressed (labeled as 1) with the help of a medical practitioner. This labeling helped the model/classifiers learn the implicit statements as well.

Table 8 states the results of all the mentioned machine learning classifiers along with all the stated techniques for feature extraction applied to the data with modified labeling.

It is evident from Table 8 that SVM outperformed all the other classifiers. SVM gave the best results with TF-IDF, uni-gram, and uni–bi-gram. With Uni–Bigram it gave the highest F1-score of 96.7. ML classifiers trained with modified labeled data were also tested with real-time tweets. Table 9 shows the results of the classifiers on the unseen data. It is evident from the results that training all the classifiers with modified labels, they performed better on the unseen data. The highest accuracy achieved was 82.3 and the highest F1-score was 82.1 by logistic regression. LR performed better on the test dataset with TF-IDF and uni–bi–tri-gram, whereas random forest gave the same highest accuracy and F1-score of 82.3 and 82.1, respectively, on test data with uni–bi-gram.

Additionally, Table 10 lists the highest accuracy and F1-scores we obtained from the raw label and modified label data when tested on the real-time tweets. Other than that, comparing the performances of the classifiers trained on raw labeled data with training on the modified labeled data, it is evident from Table 7 and Table 9 that, with the unseen/real-life data, the classifiers trained on the modified labeled data performed better. Moreover, if we consider the rate of the true negatives and true positives classification, the classifiers trained with the modified labels captured the implicit tweets more, hence fulfilling the purpose of applying machine learning in the context of depression detection.

Another aim of this study was to train the machine learning classifiers such that they has an increased rate of true positive and true negative detection whilst decreasing the rate of false positives and false negatives. Figure 3 shows the confusion matrix of TF-IDF+LR and uni–bi–tri-gram + LR; Figure 4 shows the confusion matrix of uni–bi–tri-gram + SVM. It is obvious that, in Figure 3, rate of the TPs and TNs is higher than in Figure 4 (uni–bi–tri-gram + SVM on raw-labeled-trained data).

#### 4.1.2. Ternary Classification Task with Machine Learning Classifiers

By observing the modified labeling working effectively with depression detection, we extended our work by annotating the data in another form, i.e., ternary labeling. There were many tweets in which a person tweeted about another person’s/acquaintance’s depression. In some places, the tweet provided general information about depression and the person tweeting was not actually depressed. All such tweets in which a person talked about somebody else’s depression or provided some generic information were labeled as “2” (refer to Table 3: annotation criteria). As the information is also important if somebody is talking about some other person’s depression such as his/her father, mother, or any other acquaintance, or providing some general information, this should also be considered. Therefore, to extend our work for advancement, we applied seven machine learning algorithms i.e.: K-Nearest Neighbors (KNNs), Naïve Bayes (NB), Random Forest (RF), Logistic Regression (LR), Support Vector Machine (SVM), Decision Tree (DT), and Multilayer Perceptron (MLP), on the ternary labeled data. The feature extraction techniques used were the same as those used for the binary task, i.e., TF-IDF, uni-gram, bi-gram, uni–bi-gram, and uni–bi–tri-gram.

The evaluation metrics used for assessing the performance of the classifiers were the accuracy and F1-score. The F1-score was more reliable in the ternary classification task because of the class imbalance. The number of tweets that were labeled “0” was 13,533; the number of tweets labeled “1” was 11843; the number of tweets labeled “2” was 1742 (refer to Table 4). The F1-scores are durable when classes are imbalanced and true positives and true negatives are equally important for us.

The machine learning classifiers applied with the different feature extraction methods are stated in Table 11. The experimental analysis of the ternary classification task showed that logistic regression performed the best in terms oft he F1-score with the highest score of 77.3 with the uni–bi–tri-gram method. The accuracy is also given in the table, but as the classes were imbalanced, the F1-score would be more pertinent in the case of the ternary classification. With the TF-IDF method, multilayer perceptron performed the best out of all the classifiers with an F1-score of 72.8. Logistic regression also performed well with the uni-gram method, bi-gram, and uni–bi-gram methods. In short, logistic regression performed efficiently compared to all other classifiers.

### 4.2. Experimental Analysis of Deep Learning Classification Models

For the problem of depression detection, deep learning classifiers/models also were explored and applied for the classification of depressive and non-depressive tweets. The classification models that were experimented with were feedforward networks, Convolutional Neural Networks (CNNs), Long Short-Term Memory networks (LSTM), Gated Recurrent Units (GRUs), Bi-directional LSTM (BiLSTM), and ensembles/hybrid models. Moreover, the word embedding models used were Word2Vec, FastText, and GloVe. Ultimately, a hybrid model based on the attention mechanism is presented for this study in Section 5.

#### 4.2.1. Binary Classification Task with Deep Learning and Comparative Methods

Multiple deep learning models along with word embeddings were experimented with. In Table 12, the results are shown for various combinations of word embeddings and deep learning techniques. Apart from the various combinations with word embeddings, the hybrid models/ensembles were also employed.

As a stand-alone technique, the Convolutional Neural Network (CNN) performed very well with Word2Vec with an accuracy of 95.3 and an F1-score of 95. BiLSTM also gave good results with Word2Vec with an accuracy of 94.8 and an F1-score of 94.7. Moreover, the combination of CNN+LSTM worked well as compared to CNN+BiLSTM when used with no embedding, whilst for the same ensembles when used with Word2Vec, CNN+BiLSTM performed better than CNN+LSTM. When we used FastText embedding with CNN+BiLSTM, it gave an accuracy and F1-score of 96.6.

Afterward, we also implemented the method of three-gram with CNN + LSTM and CNN +BiLSTM. CNN + LSTM performed well as compared to CNN + BiLSTM. With our data, CNN + LSTM gave an accuracy of 87.5 and an F1-score of 87.7. If we compare the related studies’ results to ours, the F1-score increased with the implemented technique. Table 13 below compares the results of our study with comparative studies [12].

Another study [12] we compared our work with gave an F1-score of 85 on the binary classification of tweets by using the skip-gram model for word representation, BiGRUs for learning latent representation of words, and CNN for classification using multimodal features, whereas, in our study, the results in terms of the F1-score improved a great deal.

Later on, with GloVe embedding, we experimented with an ensemble of LSTM + CNN and LSTM + GRU. These ensembles did not improve theF1-scores. When the GRU along with the attention layer was incorporated, the model performance greatly increased. The model’s details are elaborated in Section 5.

#### 4.2.2. Ternary Classification Task with Deep Learning Methods

In this study, deep learning models along with comparative methods were also applied to the ternary labeled data. The same combinations of word embedding with deep learning models and ensembles were used for the ternary data as used for the binary data. Table 14 presents the results of the deep learning models and ensembles on the ternary data.

On the ternary labeled data, the simple feedforward network performed relatively better on the ternary data compared to other methods such as the GRU and Word2Vec with the CNN. The simple CNN + LSTM with no embedding also gave better results with an accuracyand F1-score of 77. Moreover, FastText with BiLSTM + CNN performed very well with an accuracyand F1-score of 80.1. FastText with CNN +LSTM gave an accuracyand F1-score of 79.4. With the comparative methods [12], the results with ternary labels decreased the accuracy and F1-score, with a value of 60.1. Moreover, with GloVe, results dropped below the 70s, but when the GRU along with the attention layer was added to the model, the performance improved in terms of the accuracy and F1-score. The highest accuracy and F1-score achieved was 82.9, and the ensemble that gave this result is elaborated in the proposed model in Section 5.

## 5. Proposed Deep Learning Hybrid SSCL Classification Model Based on Attention Mechanism

Our final proposed hybrid deep learning model is based on a self-attention mechanism. In this study, we propose the SSCL classification model, i.e., the “sequence, semantic, and context learning” classification model. In our proposed model, GloVe word embedding was used for feature extraction along with a hybrid model consisting of Long Short-Term Memory (LSTM), a Convolutional Neural Network (CNN), a Gated Recurrent Unit (GRU), and an attention layer. The proposed model gave the best performance in terms of the accuracy and F1-score. Figure 5 illustrates the architecture of the proposed methodology.

Word embeddings can be categorized as learning models that help draw the representation or underlying connotation of words in our vocabulary in a semantic space that is in a lower dimension. Their fundamental principle is founded on augmenting an objective function that assists in fetching words that are frequently occurring with each other under a specific contextual window and words that occur in closer proximity in the semantic space. An extraordinary aptitude of these models is that they can efficiently capture numerous lexical properties in NLP tasks such as the similarity amongst words, correspondences between words, the global occurrence of these words in a document or a sentence, etc. These models have been developed to be increasingly prevalent in the NLP domain, and they have been utilized as an input to deep learning models. In this study, we practiced with the most popular models, but GloVe gave the best performance with our proposed model.

As stated before, *GloVe is Stanford’s* developed unsupervised learning model for discovering meaningful vector representations for words. In this study, we used GloVe’s pre-trained word embeddings, which have already been trained on a huge data corpus; thus, it can be used to solve similar NLP tasks. We used GloVe’s pre-trained vector model to map the tweets onto the lower-dimensional vector space. Twitter’s two billion tweets with 27 billion tokens and 1.2 million vocabularies with 200-dimensional vectors were used in this study to extract the textual features of the tweets to our defined size of the embedding matrix. The embedding matrix basically contains the word embeddings extracted with the help of the word embedding models.

Word embeddings from GloVe are in the form of an embedding layer with an input dimension of 5000, an input length of 120, an output dimension of 200, and an embedding matrix of (5000, 200). After, the embeddingdropout layer was deployed with the rate of 0.3. This was performed to avoid any overfitting and to avoid randomly selected neurons.

The input to our LSTM layer was the GloVe embedding matrix containing word embeddings, which not only incorporates local context information (local statistics), but also considers global word occurrence (global statistics). These word embeddings were further fed to our novel hybrid deep learning attention-based model with a learning rate of 0.01. The first layer of our proposed model consisted of LSTM with 96 LSTM cells, which are capable of acting as memory cells. LSTM contains memory cells instead of the hidden layer, which makes it good at finding long-range dependencies in the data. Long-range dependencies are crucial in finding the sentence structure. LSTMs are fundamentally made of three gates: input gate, forget gate, and output gate, which help to absorb which information should be conserved and which information should be thrown out. Moreover, LSTMs are also capable of storing input sequences in their memory cells for longer periods of time, and they fixed the problem of Recurrent Neural Networks (RNNs), where there is no regulation of the input. As RNNs do not know which past inputs need to be forgotten and which should be carried forward, LSTMs are more appropriate for the task of learning word embeddings in a semantic way and are suitable for maintaining the sequence of the word embeddings of tweets.

The output sequences of LSTM were given as the input to a 1D Convolutional Neural Network (CNN). While LSTM maintains the order/sequence of the semantics in a tweet/sentence, the CNN is capable of extracting semantic information from a sequence of sentences, and it can learn effective and suitable feature representations. The 1D CNN was used with a filter of 32 and a kernel size of five. After deploying a CNN layer, a dropout layer with a rate of 0.3 was used to avoid any overfitting from the combination of LSTM with the CNN.

Afterward, the GRU layer was deployed. While LSTM can effectively remember longer sequences than the GRU, at the same time, the GRU has better performance in capturing long-distance relationships between words in a sequence/sentence. Moreover, the GRU has proven to be computationally effective because of its precise internal structure with an update and reset gate, and it can capture long-distance relations among information. The GRU focuses on all input variables equally, which are output from its previous layer, i.e., the CNN, so it does not result in good precision/accuracy of the output sequence. As all input variables can contribute differently to the output prediction, the GRU as a stand-alone assistance technique to the LSTM+CNN cannot perform well.

Hence, in order to pay more attention to the contextual information, we used a GRU with a self-attention layer. The GRU outputs the input variables fed to it with a focus on all of the input variables, so an attention layer was deployed to focus more on the relevant words’ context. A self-attention layer was chosen instead of an attention layer because the attention mechanism only lets the output focus the attention on a particular input, whereas self-attention lets inputs of the network interact with each other, hence calculating the attention of all other inputs with respect to one input. The self-attention phenomenon is capable of drawing global dependencies between inputs and outputs. Figure 6 shows the connection of the GRU with the attention layer.

Therefore, a self-attention mechanism was deployed to focus more on the relevant features/variables from the input.

Self-attention is capable of modeling dependencies between various parts of a sequence such as understanding the syntactic function between words. Since every word in a sentence donates differently to the result detection, an attention mechanism with a GRU deployed can pay more attention to the context. The self-attention layer with the softmax function was applied to learn the contribution weights of various words in a tweet. This weight vector was multiplied with different relevant words to learn the contribution of these words in prediction and pay attention to them according to their weights. The softmax function in the self-attention layer also increases the rate of learning the semantic features.

In the end, there were three Fully Connected Layers (FCLs) applied with different Activation Functions (AF). The first FCL was constituted by a dense layer with the ReLU activation function and 32 learning units/neurons. After that, the second FCL also had a ReLU activation function with eight neurons. Lastly, an FCL with a sigmoid activation function was used. The sigmoid function is useful in that it can be used in classification tasks, due to the fact that the output value is scaled to the range of 0 and 1. Moreover, the sigmoid function can be used for binary, as well as for multi-label classification.

In our proposed model, the “Adam” optimizer was used in combination with the “binary cross-entropy” loss function. Figure 7 elaborates an example of a tweet fed to our proposed model and illustrates in detail that that part of our proposed model extracts the sequence, sentiment, and context vector from the tweet. In Section 6, the table represents the results of the proposed model with the presented binary and ternary labeled data.

## 6. Results and Discussion

This section discusses the results obtained from the proposed deep learning hybrid model based on the attention mechanism, and it compares the proposed model with other methods that have been employed for the task. In Section 4.2.1 and Section 4.2.2, the experimental analysis of deep learning models with binary and ternary labeled data was presented, respectively. The depression classification task was divided into two categories of binary and ternary classification, as already mentioned in the above sections. Each task was experimented with explicitly by applying various feature extraction methods with machine learning and deep learning algorithms. For the binary classification task, logistic regression and SVM gave the best accuracy with all of the feature extraction techniques except bi-grams. With bi-grams, MLP performed well (refer to Table 6). For the performance evaluation of the classifiers, unseen data were used to test the models against modified labels and raw labeled data. It is clearly evident in Table 7 and Table 9 that, with the modified labels, the performance of the classifiers was better as compared to the raw labeled data. For the raw-labeled-trained models, the highest F1-score was obtained by LR and SVM with the uni–bi–tri-gram technique. The highest F1-scores of 76.3 and 76.4 were for LR and SVM, respectively, on the unseen data. Furthermore, with the modified-labeled-trained models, the unseen data were also tested. The highest F1-score was obtained by LR with uni–bi–tri-gram on the unseen data, and the score was 82.3. If we compare the scores of the models trained with raw labels and modified labels, there was a huge difference in terms of the accuracy and F1-score. Hence, implicit depression detection and an increase in the number of true positives and true negatives increased.

Deep learning models were also employed for the depression detection task. Referring to Table 12, it can be seen that the combination of the CNN+LSTM worked well as compared to the CNN+BiLSTM when used with no embedding, whilst the same ensembles when used with Word2Vec and the CNN+BiLSTM performed better than the CNN+LSTM. When we used FastText embedding with the CNN+BiLSTM, it gave an accuracy and F1-score of 96.6. With our proposed model that used GloVe 200-d with the LSTM+CNN+GRU +attention, the F1-score increased to 97.4 percent.

Similarly, all of the machine learning algorithms that were experimented with for binary labels were also applied for ternary labeled data. In the case of ternary labels, LR performed best with the uni–bi–tri gram technique and gave an accuracy of 91.4 and an F1-score of 77.3 (refer to Table 10). When the deep learning models were experimented with, the simple feedforward network performed relatively better on the ternary data as compared to other methods such as the GRU and Word2Vec with the CNN (refer to Table 14). The BiLSTM + CNN performed very well with an accuracy and F1-score of 80.1. With GloVe, the results dropped below the 70s, but when the GRU along with the attention layer was added to the model, the performance was improved in terms of the accuracy and F1-score. The highest accuracy and F1-score achieved was 82.9, and the ensemble that gave this result was our proposed model.

To achieve better results for the task of depression classification in terms of implicit and explicit depression detection, we proposed a model in which, along with an ensemble of the CNN and LSTM, we used the GRU and self-attention mechanism to pay attention to the contextual information of the text also. In other methods, the focus was on extracting the sequential and semantic information, whereas, in our proposed model, LSTM acted as a sequence learning mechanism and the CNN paid attention to the semantics and significant features. To further enhance the performance of implicit depression detection we added the GRU along with a self-attention layer to grasp the contextual information that a text carries. The GRU captured long-distance relationships between the occurrence of words, whereas the self-attention layer focused on relevant and pertinent features, hence improving the performance in terms of the accuracy and F1-score, as well as enhancing the performance in terms of implicit depression detection tasks and increasing the number of true positive and true negative results.

Table 15 presents the improved results with the proposed model. It is evident by seeing the experimental results of the binary and ternary labeled data and the results in Table 15 that our model improved the F1-score and accuracy. Our model was implemented with GloVe word embedding and, afterward, also tested with FastText. Our model performed better with GloVe. On the unseen test data, our model gave an accuracy and F1-score of 94.4 in the case of binary labeled data. Moreover, if we observe our results as compared to studies included in the related work, the proposed model improved the accuracy of binary labeled data a great deal, as shown in Table 13.

We also implemented the comparative methods listed in Table 13 with our ternary data. The three-gram with CNN + LSTM gave an accuracy of 60.1 and an F1-score of 60, whereas, with our deep-learning-based method, the accuracy and F1-score improved for ternary labeled data also, with an accuracy and F1-score of 82.9.

Figure 8 depicts the validation accuracy graph for binary labeled data, which increased with learning. The loss decreased as the model’s learning increased.

Figure 9 illustrates the accuracy curve for ternary labeled data.

### Cross-Domain Validation of Proposed Model

The proposed model was validated through cross-domain validation by applying it to sarcasm detection in news headlines. The dataset that was used is available publicly on Kaggle [35]. The dataset was from theonion.com and huffingtonpost.com, and it consisted of sarcastic and non-sarcastic news headlines. Table 16 lists the details. The same preprocessing techniques used for the depression detection dataset were applied to the sarcasm detection dataset, and the proposed model was also applied to the sarcasm dataset.

Table 17 presents the results of the comparative model from a few studies based on the same dataset, along with our model’s result on the sarcasm detection dataset. Our model improved the accuracy and F1-score of sarcasm detection based on the news headlines data.

Figure 10 below depicts the validation accuracy, in the case of sarcasm detection through our proposed model.

## 7. Conclusions and Future Work

This research mainly focused on the data annotation of tweets and explored various feature extraction techniques along with machine learning and deep learning models. Ensembles were also employed for the task of depression detection. Data annotation was performed in two ways, i.e., binary labeling of data and ternary labeling of data. In the binary annotation, the tweets were labeled as depressed and non-depressed, whereas, in the ternary labeling of the data, tweets were labeled as depressed, non-depressed, and tweets about another person’s depression or some other information about depression. Afterwards, data preprocessing specific to our dataset was performed in order to improve the learning of the models. We then applied machine learning and deep learning algorithms to both types of annotations discretely. Ultimately, a hybrid deep learning framework based on an attention mechanism was proposed. SVM performed best with the uni–bi-gram technique for the binary labeled data with an accuracy of 96.8 and an F1-score of 96.7. For the ternary labeled data, logistic regression performed best with an accuracy of 91.4 and an F1-score of 77.3. With deep learning models, we experimented with various models and ensembles, of which FastText with BiLSTM and the CNN performed well for binary data with an accuracy an F1-score of 96.6, whereas our proposed framework further improved the accuracy and F1-score to 97.4. For deep learning when applied on the ternary labels, the same experiment with FastText along with BiLSTM and the CNN gave an accuracy and F1-score of 80.1. Our proposed framework further improved the performance on the ternary labeled data and gave an accuracy and F1-score of 82.9. Our proposed model was also cross-validated in the sarcasm detection domain. In the cross-validation, the accuracy and F1 score of the previous studies were also improved by our proposed framework.

In the future, we will aim to extend our work in two ways. We would like to train our model/framework on balanced ternary labeled data to improve its performance in terms of the accuracy and F1-score. Multiclassification is important for us because, in this way, we can capture the actual context and subject of the tweet. Furthermore, depression severity in the form of scores can be included in the studies to gauge the gravity of the disease.

## Figures and Tables

**Figure 1 sensors-22-09775-f001:**
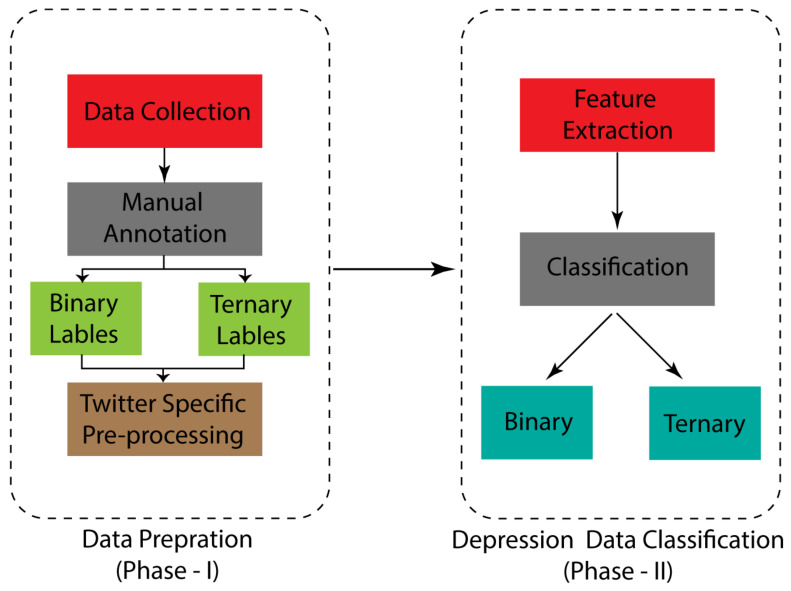
Phases of the depression classification research.

**Figure 2 sensors-22-09775-f002:**
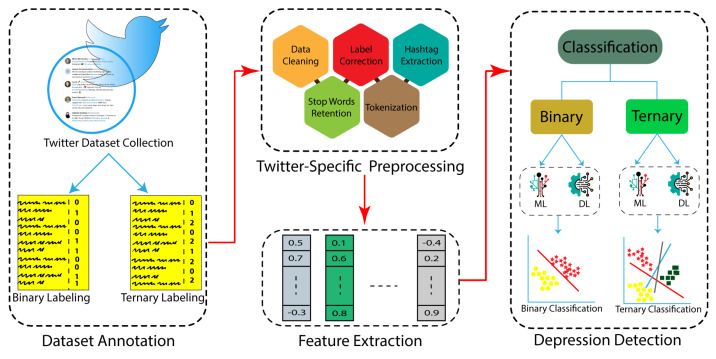
Methodological summary of the depression classification task.

**Figure 3 sensors-22-09775-f003:**
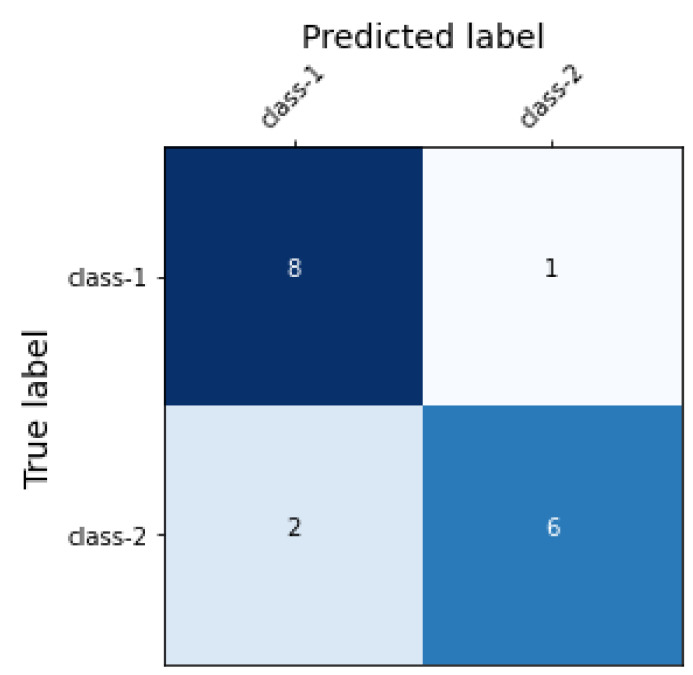
Confusion matrix of TF-IDF+LR and uni–bi–tri-gram + LR.

**Figure 4 sensors-22-09775-f004:**
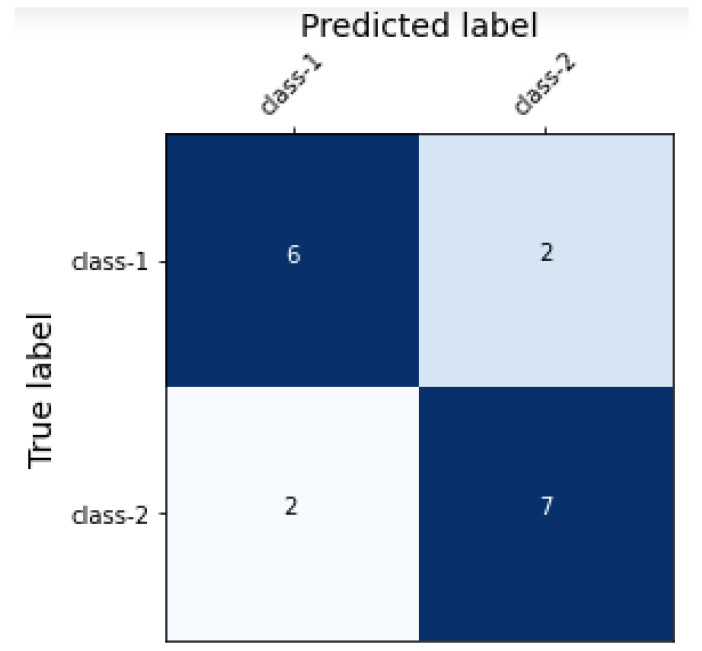
Confusion matrix of uni–bi–tri-gram + SVM.

**Figure 5 sensors-22-09775-f005:**
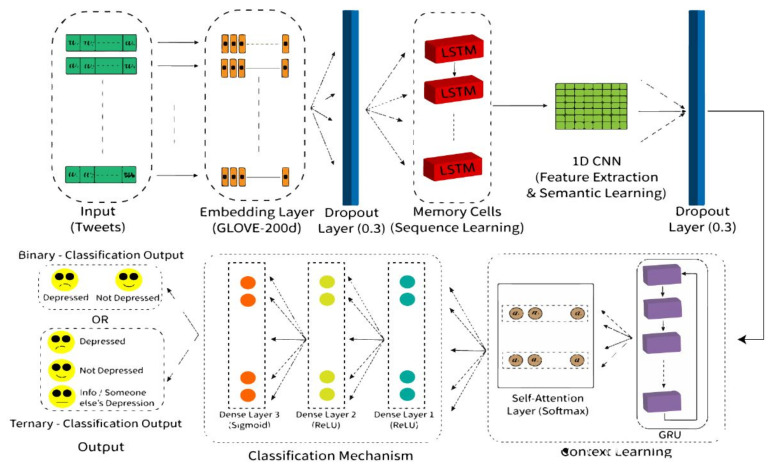
Architectural model of proposed methodology.

**Figure 6 sensors-22-09775-f006:**
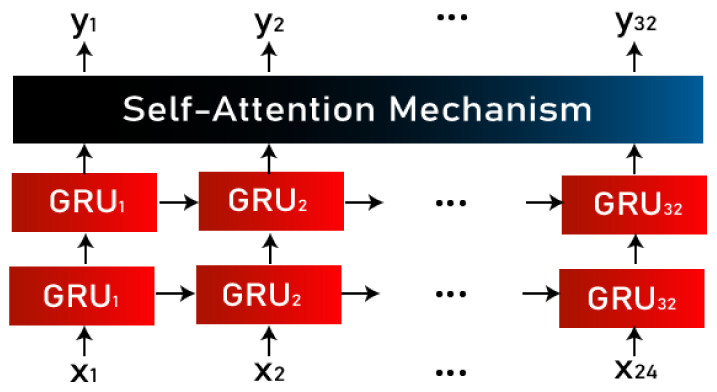
Connection of GRU with the attention layer.

**Figure 7 sensors-22-09775-f007:**
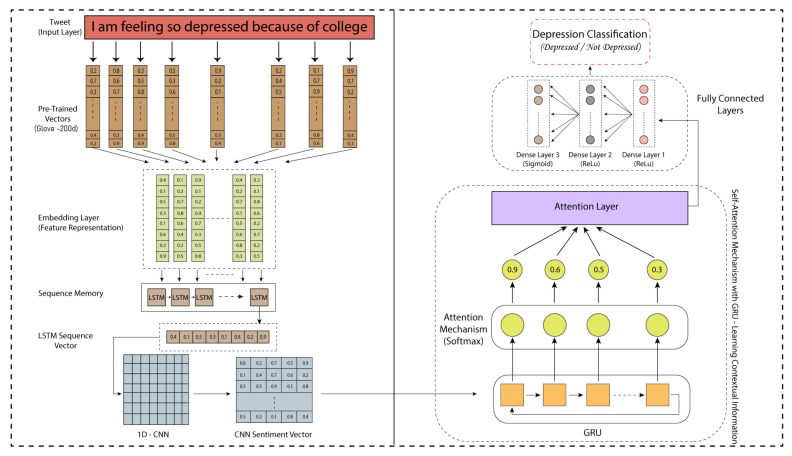
An example tweet processed through our proposed framework.

**Figure 8 sensors-22-09775-f008:**
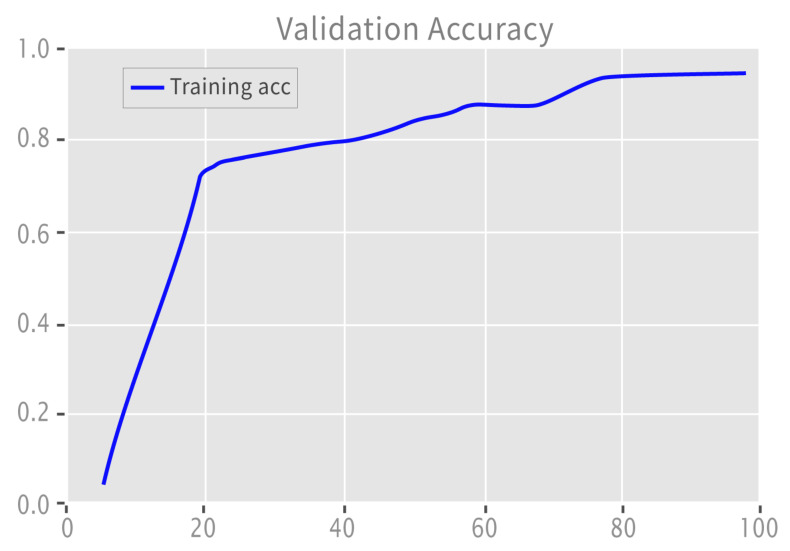
Validation accuracy: binary.

**Figure 9 sensors-22-09775-f009:**
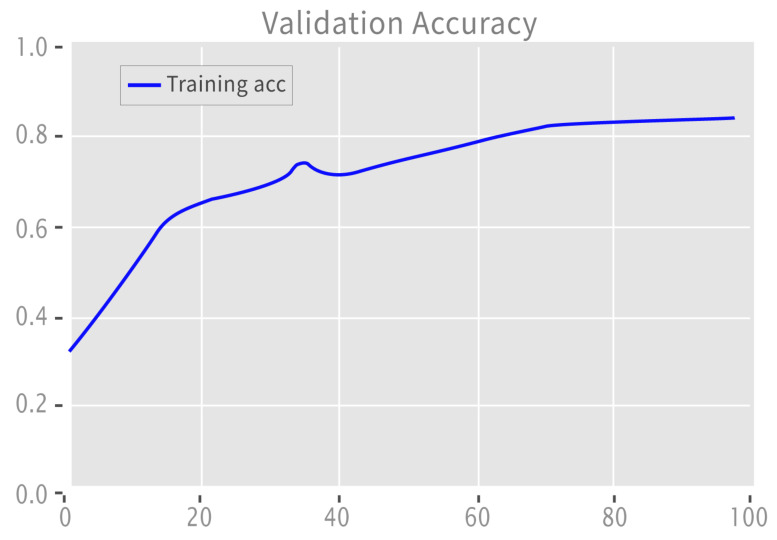
Validation accuracy: ternary.

**Figure 10 sensors-22-09775-f010:**
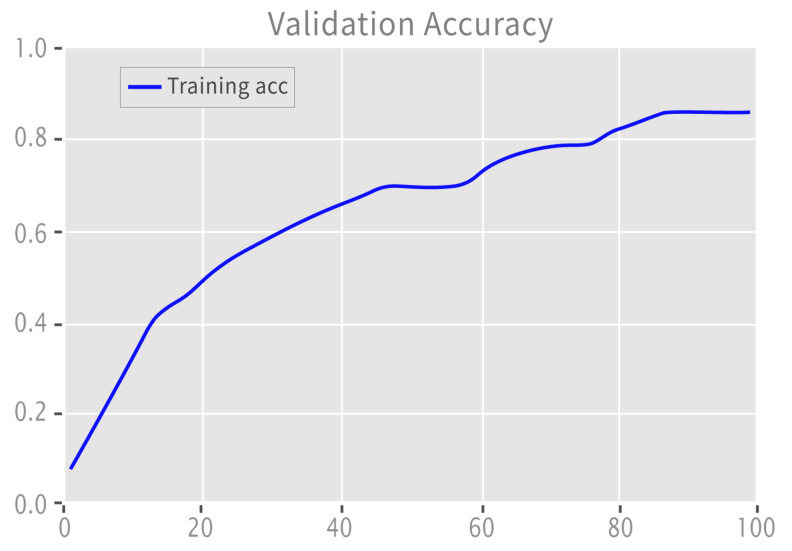
Validation accuracy: SD.

**Table 1 sensors-22-09775-t001:** Elaboration of TPs, TNs, FPs, and FNs.

True Positive (TP)	Depression instances that are positive and determined as positive
True Negative (TN)	Instances that are negative and determined as negative/not-depressed
False Positives (FP)	Instances that are negative and determined as positive/depressed
False Negative (FN)	Depression instances that are positive, but are determined as negative

**Table 2 sensors-22-09775-t002:** Elaboration of TPs, TNs, FPs, and FNs with samples.

	Sample of the Labels 0 and 1
FP (labeled “1”, actually “0”)	Damn louis really did pull me out of my depression painting for the first time in months. Damn you ever gotten so good that you cured your depression Cleaned my room yesterday and shit good damn mess already
FN (labeled “0”, actually “1”)	Think your life is hard you have to wake up and hide three chronic illnesses secret I have done lot of reading on what’s going on in Nigeria haven’t been able to donate because I am jobless and my worries don’t let me sleep. Sometimes my brain literally can’t comprehend how life is real like the fact it does exist

**Table 3 sensors-22-09775-t003:** Annotation criteria.

Label Name in File	Description
Label	The initial label of the file with false positives and false negatives. Depression negation tweets (in which the person has stated that he/she is not depressed) and other tweets were labeled on the criteria of particular keyword existence and not on the actual context.
Binary label	The binary label is a modified labeling of the file, which defines the classification problem into 2 class labels, “0” and “1”.
	**Label “0”:** Tweets that are not depicting depression The tweets with negation, e.g., “not depressed” Tweets with solved depression problem, e.g., “my depression is cured” **Label “1”:** Tweets that are depicting depression. Depression can be of some person who is talking of himself/herself. A tweet that refers to another person’s depression. A tweet with some important information of depression.
Ternary label (multi label)	Multi-class labeling is another variation of modified labels consisting of 0, 1, and 2
	**Label “0”:** Tweets that are not depicting depression The tweets with negation, e.g., “not depressed” Tweets with solved depression problem, e.g., “my depression is cured”
	**Label “1”:** Tweets that are depicting depression. Depression can be of some person who is talking of himself/herself.
	**Label “2”:** A tweet that refers to another person’s depression. A tweet with some important information of depression.

**Table 4 sensors-22-09775-t004:** Number of labels.

Raw Tweets	Number of 0s: 15,000
	Number of 1s: 15,999
	Number of FPs: 930
	Number of FNs: 39
Binary Labeling	Number of 0s: 13,533
	Number of 1s: 13,584
Ternary Labeling	Number of 0s: 13,533
	Number of 1s: 11,843
	Number of 2s: 1742

**Table 5 sensors-22-09775-t005:** Elaboration of N-grams.

N-Grams	Tweet: “My life has become difficult”
Uni-Gram (N = 1)	My, life, has, become, difficult
Bi-Gram (N = 2)	My life, life has, has become, become difficult
Uni–Bi-Gram (N = 1, 2)	My, life, has, become, difficult, my life, life has, has become, become difficult
Uni–Bi–Tri-Gram (N = 1, 2, 3)	My, life, has, become, difficult, my life, life has, has become, become difficult, My life has, life has become, has become difficult, etc.

**Table 6 sensors-22-09775-t006:** Results of machine learning classifiers with raw binary labels.

Traditional Features										
**Techniques**	**TF-IDF**		**Uni-Gram**		**Bi-Gram**		**Uni–Bi-Gram**		**Uni–Bi–Tri-Gram**	
	**Accuracy**	**F1-Score**	**Accuracy**	**F1-Score**	**Accuracy**	**F1-Score**	**Accuracy**	**F1-Score**	**Accuracy**	**F1-Score**
**KNN**	82.1	81.6	94	93.9	75.9	75.5	92.8	92.8	91.9	91.9
**NB**	81.7	81	92	91.9	89.3	89.1	89	88.8	89.6	89.4
**RF**	98.6	98.5	99	99	95.8	95.8	98.6	98.6	98.5	98.5
**LR**	99.1	99	99.8	99.8	95.8	95.8	99.8	99.8	99.7	99.7
**SVM**	99.6	99.6	99.8	99.8	95.4	95.4	99.7	99.7	99.6	99.6
**DT**	99	98.9	99.6	99.6	95	95	99.3	99.3	99.4	99.4
**MLP**	98.7	98.7	99.6	99.6	97.4	97.3	99	99	98.7	98.7

**Table 7 sensors-22-09775-t007:** Real-time tweets dataset result for ML classifiers with raw labeling.

Traditional Features										
**Techniques**	**TF-IDF**		**Uni-Gram**		**Bi-Gram**		**Uni–Bi-Gram**		**Uni–Bi–Tri-Gram**	
	**Accuracy**	**F1-Score**	**Accuracy**	**F1-Score**	**Accuracy**	**F1-Score**	**Accuracy**	**F1-Score**	**Accuracy**	**F1-Score**
**KNN**	47	43.9	70.5	70.1	35.5	34.3	70.5	70.1	70.5	70.5
**NB**	47	31.9	52.9	48.4	47	31.9	47	31.9	47	31.9
**RF**	64.7	61.3	75.4	75.7	70.5	70.1	70.5	68.8	58.8	56.4
**LR**	70.5	70.1	64.7	63.5	70.5	70.1	70.5	70.1	76.4	76.3
**SVM**	64.7	63.5	64.7	63.5	70.5	70.5	75.4	76.3	76.4	76.4
**DT**	58.8	56.4	58.8	56.4	70.8	68.8	58.8	56.4	52.9	48.4
**MLP**	58.8	58.2	58.8	58.2	64.7	61.3	58.8	58.2	64.7	64.5

**Table 8 sensors-22-09775-t008:** Machine learning classifiers applied on modified labeled data.

Traditional Features										
**Techniques**	**TF-IDF**		**Uni-Gram**		**Bi-Gram**		**Uni–Bi-Gram**		**Uni–Bi–Tri-Gram**	
	**Accuracy**	**F1-Score**	**Accuracy**	**F1-Score**	**Accuracy**	**F1-Score**	**Accuracy**	**F1-Score**	**Accuracy**	**F1-Score**
**KNN**	80.2	79.8	82.8	82.6	63.1	57.7	84.8	84.7	59.9	56.1
**NB**	86.9	86.7	92.6	92.5	89	88.9	90.8	90.7	90	89.8
**RF**	96.1	96	96.4	96.2	93.9	93.6	96.4	96.2	95.9	95.8
**LR**	95.3	95.2	96.5	96.3	93.5	93.3	96.9	96.7	96.6	96.5
**SVM**	96.4	96.3	96.6	96.5	92.7	92.6	96.8	96.7	96.5	96.4
**DT**	95.3	95	96	95.8	92.3	92.2	96.3	96.2	96.3	96.1
**MLP**	94.7	94.3	95.3	95	93.9	93.7	95.7	95.5	91.8	91.7

**Table 9 sensors-22-09775-t009:** Real-time tweets dataset result for ML classifiers with modified labeling.

Traditional Features										
**Techniques**	**TF-IDF**		**Uni-Gram**		**Bi-Gram**		**Uni–Bi-Gram**		**Uni–Bi–Tri-Gram**	
	**Accuracy**	**F1-Score**	**Accuracy**	**F1-Score**	**Accuracy**	**F1-Score**	**Accuracy**	**F1-Score**	**Accuracy**	**F1-Score**
**KNN**	70.5	70.1	64.7	64.5	58.8	47.1	58.8	58.2	70.5	70.1
**NB**	58.8	52.9	64.7	63.5	52.9	43.3	64.7	61.3	58.8	52.9
**RF**	76.4	75.7	64.7	63.5	76.4	76.3	82.3	82.1	64.7	63.5
**LR**	82.3	82.1	70.5	70.4	64.7	63.5	70.5	70.1	82.3	82.3
**SVM**	70.5	70.1	70.5	70.1	70.5	70.1	76.4	76.3	76.4	76.3
**DT**	70.5	70.4	64.7	63.5	70.5	70.1	64.7	64.5	58.8	56.4
**MLP**	76.4	75.7	70.5	70.1	52.9	52.7	64.7	63.5	76.4	76.3

**Table 10 sensors-22-09775-t010:** Machine learning classifiers applied on ternary labeled data.

Traditional Features										
**Techniques**	**TF-IDF**		**Uni-Gram**		**Bi-Gram**		**Uni–Bi-Gram**		**Uni–Bi–Tri-Gram**	
	**Accuracy**	**F1-Score**	**Accuracy**	**F1-Score**	**Accuracy**	**F1-Score**	**Accuracy**	**F1-Score**	**Accuracy**	**F1-Score**
**KNN**	78	59.6	84.3	61.4	50.8	34.8	80	55.5	66.8	45.5
**NB**	84.8	58.6	86.5	60.4	83.7	59	86.3	59.8	86	59.9
**RF**	89.7	62.4	90	63.1	88.1	61.5	90.5	63.2	89.7	62.6
**LR**	90.4	72.7	90.8	77.1	88.1	70.3	91.5	77.1	91.4	77.3
**SVM**	90.8	71.3	91.2	72.8	86.7	59.9	91.5	70.2	90.3	67.7
**DT**	88	72.1	88.3	72	85.2	66.3	89.4	73.2	87.9	72.1
**MLP**	88.7	72.8	89.6	75.3	88.7	69.7	89.4	72.9	88.8	74.2

**Table 11 sensors-22-09775-t011:** Comparison of highest accuracy and F1-score on real-time tweets.

Label	Techniques	Accuracy	F1-Score
Modified-Labeled-Trained Data	TF-IDF + LR	82.3	82.1
	Uni–bi–tri-gram + LR	82.3	82.1
Raw-Labeled-Trained Data	Uni–bi–tri-gram + SVM	76.4	76.4

**Table 12 sensors-22-09775-t012:** Experimental analysis of deep learning models on binary labeled Data Section 4.2.1.

Embeddings	Technique	Accuracy	F1- Score
None	Feedforward Network	89.1	89
None	GRU	49.8	49.8
Word2Vec	GRU	48.5	48.4
Word2Vec	CNN	95.3	95
Word2Vec	LSTM	91.8	91.5
Word2Vec	BiLSTM	94.8	94.7
None	CNN + BLSTM	95.2	95.2
None	CNN + LSTM	95.5	95.5
Word2Vec	CNN + BLSTM	96.1	96.1
Word2Vec	CNN + LSTM	95.9	95.9
3-Gram	LSTM	84.9	82.7
3-Gram	CNN + BLSTM	87.2	87
3-Gram	CNN + LSTM	87.5	87.7
FastText	BiLSTM + CNN	96.6	96.6
FastText	LSTM + CNN	96.2	96.2
GloVe	LSTM	60	60
GloVe	CNN	74	74
GloVe	LSTM + CNN	75	75
GloVe	LSTM + GRU	61	61
GloVe	LSTM + CNN + GRU	74.7	74.7
FastText	CNN + LSTM + GRU +Attention	96.7	96.7
GloVe	LSTM + CNN + GRU +Attention	97.4	97.4

**Table 13 sensors-22-09775-t013:** Comparison of the methods applied on our data.

Technique	Results with Our Dataset	
	**Accuracy**	**F1-Score**
3-Gram + CNN + BiLSTM	87.2	87
3-Gram + CNN + LSTM	87.5	87.7

**Table 14 sensors-22-09775-t014:** Experimental analysis of deep learning models on ternary labeled data.

Embeddings	Technique	Accuracy	F1 Score
None	Feedforward Network	77.4	77
None	GRU	42.1	42.1
Word2Vec	GRU	43	43
Word2Vec	CNN	76.3	76.2
Word2Vec	LSTM	75.5	75.5
Word2Vec	BiLSTM	72.5	72.5
None	CNN + BLSTM	63	63
None	CNN + LSTM	77	77
Word2Vec	CNN + BLSTM	63.9	63.9
Word2Vec	CNN + LSTM	64	64
3-Gram	LSTM	54.3	54.3
3-Gram	CNN + BLSTM	60	60
3-Gram	CNN + LSTM	60.1	60
FastText	BiLSTM + CNN	80.1	80.1
FastText	LSTM + CNN	79.4	79.4
GloVe	LSTM	52.1	52.1
GloVe	CNN	69.2	69.2
GloVe	LSTM + CNN	68.6	68.6
GloVe	LSTM + GRU	55.1	55.1
GloVe	LSTM + CNN + GRU	71.9	71.9
FastText	CNN + LSTM + GRU +Attention	81.9	81.9
GloVe	LSTM + CNN + GRU +Attention	82.9	82.9

**Table 15 sensors-22-09775-t015:** Proposed method results on binary and ternary labeled data.

Label	Embeddings	Technique	Accuracy	F1-Score
Binary Data	GloVe	LSTM + CNN + GRU +Attention	97.4	97.4
	FastText	CNN + LSTM + GRU +Attention	96.7	96.7
Ternary Data	GloVe	LSTM + CNN + GRU +Attention	82.9	82.9
	FastText	CNN + LSTM + GRU +Attention	81.9	81.9

**Table 16 sensors-22-09775-t016:** News headline sarcasm detection dataset details.

Total headlines	26,709
Sarcastic headlines	11,723
Non-sarcastic headlines	14,984

**Table 17 sensors-22-09775-t017:** Comparative methods’ result and our model’s result.

Research Paper	Explanation	Results Claimed
Optimal Feature Extraction based Machine Learning Approach for Sarcasm Type Detection in News Headlines [36]	The study focused on optimal feature extraction using skip-gram along with SVM.	Accuracy 78.82%, F1-score 76.42%
Identification of Sarcasm in Textual Data: A Comparative Study [37]	Word embeddings such as GloVe and FastText were used. Highest accuracy with the GloVe + LSTM + CNN.	Accuracy 81.6%, F1-score NA
Deep CNN-LSTM with Word Embeddings for News Headline Sarcasm Detection [38]	Embedding vector used for capturing the words in the headlines along with the CNN + BiLSTM	Accuracy 86.1%, F1-score NA
Our Study	GloVe + LSTM + CNN + GRU + attention layer	Accuracy 86.7%, F1-score 86.7%

## Data Availability

Not applicable.

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
