# Peer review of "Depression Detection Based on Hybrid Deep Learning SSCL Framework Using Self-Attention Mechanism: An Application to Social Networking Data"

_sensors, 2022, doi:10.3390/s22249775_

Round 1

Reviewer 1 Report

This manuscript proposes a novel scheme for depression detection from the textual data. the solved problems are specific and meaningful in today's society. My suggestion is acceptance after minor revision. While this paper still has many spelling and grammarical errors. Formats of many places should be checked carefully. Here are my some suggestions:

1. The proposed new scheme should have a specific name or an abbreviated title.

2. There are some little errors like 'paying 'attention' at the third line and 'False positives and False Negatives' at the sixed line of the listed pivotal contributions, etc.

3. It should be consistent that the present tense is used at the beginning of the third paragraph and the past tense is used at the beginning of the fourth paragraph in Related Work.

4. Table 1. and 2. are too narrow for the description words to spread. 

5. The section of methodological overview and datasets preparation in Chapter 3 are not clearly delineated.

6. There are many places in this paper that can be streamlined, i.e., the concepts of binary classification, ternary classification and F1-score are mentioned too many times; Chapter 4 describes too much about the experimental analysis of traditional machine learning classification models, etc.

7. Figure 6. is too roughly designed and does not have clear figure notes.

8. The words desciption of figure 6. do not highlight the superiority of the model and are not expressed clearly.

9. Some advanced and recently developed techniques in deep learning should be reviewed or compared such as:A self-adaptive mutation neural architecture search algorithm based on blocks,Partial connection based on channel attention for differentiable neural architecture search,A multiobjective evolutionary approach based on graph-in-graph for neural architecture search of convolutional neural networks

10. The paper should highlight the most novel part and remove that not matters.

Author Response

All the comments of Reviewer 1 have been answered in a word file (Attached) 

Reviewer 2 Report

The paper discusses a topical issue in computer science which, however, seems to be outside the thematic scope of the Sensors journal and in particular from that of the Special issue “3D Sensing and Imaging for Biomedical Investigations”. On the other hand, the increasing growth of mental health diseases in developed countries and their great social importance make it highly recommended to publish research results related to the timely detection of depressive conditions.

The presentation of the paper is well organized and written at a good professional level. The used research methodology is suitable for the solved task. Some promising results in the field of study are described at a sufficient level of detailing. However, the presented results should necessarily be interpreted by experts in the field of psychiatry. Also, in this case, it should be considered to combine the methods used by other AI/NLP methods which allow the generation of meaningful explanations of the obtained results.

Author Response

All the comments of Reviewer 2 have been answered in a word file (Attached) 
